# A novel mean shape based post-processing method for enhancing deep learning lower-limb muscle segmentation accuracy

Zhicheng Lin[1], Enrico Dall'Ara[2,3], Lingzhong Guo[1,3]*

1 Department of Automatic Control and Systems Engineering, University of Sheffield, Sheffield, United Kingdom, 2 Division of Clinical Medicine, University of Sheffield, Sheffield, United Kingdom, 3 Insigneo, University of Sheffield, Sheffield, United Kingdom

* l.guo@sheffield.ac.uk

**Data Availability Statement:** The data from two different cohorts of healthy post-menopausal women (PMW) were used in the study. The first cohort (PMW-1) includes 10 PMW T1-weighted

## Abstract

This study aims at improving the lower-limb muscle segmentation accuracy of deep learning approaches based on Magnetic Resonance Imaging (MRI) scans, crucial for the diagnostic and therapeutic processes in musculoskeletal diseases. In general, segmentation methods such as U-Net deep learning neural networks can achieve good Dice Similarity Coefficient (DSC) values, e.g. around 0.83 to 0.91 on various cohorts. Some generic post-processing strategies have been studied to incorporate connectivity constraints into the resulting masks for the purpose of further improving the segmentation accuracy. In this paper, a novel mean shape (MS) based post-processing method is proposed, utilizing Statistical Shape Modelling (SSM) to fine-tune the segmentation output, taking into consideration the muscle anatomical shape. The methodology was compared to existing post-processing techniques and a commercial semi-automatic tool on MRI scans from two cohorts of post-menopausal women (10 Training, 8 Testing, voxel size 1.0x1.0x1.0 mm$^3$). The MS based method obtained a mean DSC of 0.83 across the different analysed muscles and the best performance for the Hausdorff Distance (HD, 20.6 mm) and the Average Symmetric Surface Distance (ASSD, 2.1 mm). These findings highlight the feasibility and potential of using anatomical mean shape in post-processing of human lower-limb muscle segmentation task and indicate that the proposed method can be popularized to other biological organ segmentation mission.

## 1 Introduction

The segmentation of human muscles in medical imaging, particularly from Magnetic Resonance Imaging (MRI) and Computed Tomography (CT) scans, represents a critical challenge in diagnostic and therapeutic processes. Some classical deep learning (DL) segmentation models such as U-Net [1] have been used to segment the lower limb muscles of children with cerebral palsy [2] and athletes [3], achieving accuracy (Dice Similarity Coefficient, DSC) around 0.87 and 0.91, respectively, and a modified U-Net [4] and a novel Spatial Channel U-Net [5] achieved DSC around 0.83–0.84 on post-menopausal women. While DL has markedly

magnetic resonance images with no muscle disease and were recruited by the Metabolic Bone Centre (Sheffield, UK) as part of larger projects (approved by the East of England–Cambridgeshire and Hertfordshire Research Ethics 126 Committee and the Health Research Authority, Reference 16/EE/0049). The second (PMW-2) was recruited from a previous observational study (approved by the Leeds West Research 128 Ethics Committee, Reference 20/YH/0274) and consisted of 8 healthy PWM without muscle diseases. All data is used anonymously. Data cannot be shared publicly for confidentiality reasons, according to the terms of the Leeds IRB. However, the interested reader can contact info@insigneo.org or the corresponding author to request access to the data. Requests for data may be granted to eligible parties on reasonable request and with the completion of any required prerequisites, such as a Data Use agreement.

**Funding:** The study was funded by Engineering and Physical Sciences Research Council (EPSRC, Frontier Multisim Grant, EP/K03877X/1, and EP/S032940/1), and the Medical Research Council (MRC) and Versus Arthritis (Medical Research Council Versus Arthritis Centre for Integrated Research into Musculoskeletal Ageing, CIMA, (MR/R502182/1) The funders had no role in study design, data collection and analysis, decision to publish, or preparation of the manuscript.

**Competing interests:** The authors have declared that no competing interests exist.

advanced this field [2, 3], the important role of post-processing in refining segmentation results cannot be neglected. Post-processing techniques allow for further modification, fixing some obvious mis-segmentation errors such as voids and boundary notches, removing unnecessary disconnected small areas, and preventing leakage, resulting in cleaner and more precise muscle boundaries [6, 7].

In muscle segmentation post-processing, it is crucial to consider muscle anatomical area, volume, and age-related regional muscle loss. Innovative techniques should focus on accurately measuring muscle average area essential for studying how muscle mass affects muscle force changes with age and training [8]. Our research is centred on this pivotal aspect, aiming to enhance the accuracy of lower limb muscle segmentation by developing innovative post-processing techniques.

In the field of medical imaging, particularly in muscle segmentation, post-processing stands as a crucial step to refine and validate the results obtained from initial segmentation algorithms (e.g., convolutional neural networks (CNNs)). In a study [9] on the automatic segmentation of lumbar paraspinal muscles, the impact of CNN architecture and training was evaluated. The study emphasized the use of post-processing transformations, such as spatial connection and closing analysis, to reduce false positives and further improve CNN accuracy. Another study [10] focused on multi-muscle deep learning segmentation for the quantification of muscle fat infiltration in cervical spine conditions. The study reported high accuracy and reliability for the CNN model and highlighted the use of post-processing techniques to minimize bias and improve segmentation accuracy. Furthermore, research on transfer learning for data-efficient abdominal muscle segmentation with CNNs emphasized the potential of transfer learning to produce data-efficient skeletal muscle segmentation models, highlighting the importance of leveraging advanced techniques to improve the efficiency of muscle segmentation [11]. For lower-limb muscle segmentation task, some morphological operations like hole-filling [3] and binary closing [2] methods were utilized to fill the empty holes or remove the outliers. In summary, post-processing techniques play a crucial role in refining and validating muscle segmentation results obtained from CNNs, ultimately improving the accuracy and efficiency of medical imaging analysis. This process is not merely a technical afterthought but a pivotal component that significantly impacts the overall accuracy and clinical utility of segmentation. The traditional post-processing techniques based on pixel grey values like thresholding and region growing, which are straightforward and effective in certain natural images, often fall short in dealing with the complexity of muscle tissues, struggling to differentiate between muscle groups and adjacent structures with similar densities or intensities.

In recent years, some new post-processing methods for enhancing image segmentation have been developed. For example, in a task of liver segmentation [6], assuming an organ should be a single volume with the maximum sized label region, a 2D post-processing method based on erosion operation, identification, and selection of the largest region, dilation operation, and edge smoothing was utilized to achieve a 4% percentage gain of Intersection over Union (IoU). The first three steps were based on the process of the normal images and operated in MATLAB (function: imerode, bwlabel, and imdilate). Edge smoothing was achieved through a 2D convolution with a uniform square filter applied to the binary labelled mask. This 2D post-processing algorithm was significant for its ability to refine liver segmentation outputs, particularly in isolating the main liver region from extraneous segments and smoothing the edges for a more accurate representation of the liver's shape and boundaries. Another example is the Conditional Random Field (CRF) post-processing methods, as presented in [12] and [13]. It represents a significant advancement in the field of image segmentation, particularly in refining the segmentation results of Deep Convolutional Neural Networks (DCNNs). CRF was originally attached to the end of DeepLab V1 [12], and achieved 4%

improvement of IoU. The CRF in DeepLab V1 focused on recovering detailed local structure, instead of further smoothing the already smooth outputs from modern DCNNs. The DenseCRF proposed in [13] presents an efficient inference algorithm for fully connected CRFs, increasing the performance of the unary classifiers on the VOC 2010 from 13% to 22% accuracy.

Although they have achieved commendable segmentation benefits, previous studies did not consider the 2D or 3D shape of the segmentation targets. This is particularly relevant for muscles, which, despite varying in volume across different subjects, share similarities in spatial morphology and topological structure [14, 15]. The effectiveness of CRF and 2D post-processing methods depends considerably on the performance of the underlying CNN models; if the models do not achieve accurate results or good feature maps, the post-processing techniques will not be likely to yield significant improvements. Incorporating mean shapes or representative shapes for optimizing the segmentation results can enable the inclusion of the anatomical significance of the segmentation targets or in biomedical modelling, leading to potentially more accurate results, such as in bone segmentation [16] and in human lower-limb anatomy modelling [17]. In this study, we propose a new post-processing approach based on the mean shape (MS) and a Statistical Shape Model (SSM) [18] to further refine deep learning-based muscle segmentation results, with a focus on accurately delineating muscle boundaries and reducing false positives.

The goal of this study was to propose a new mean shape based post-processing method and to assess the effect of it with existing methods (CRF, and 2D post-processing method) on the accuracy of a previously developed CNN approach to segment individual muscles of the lower limb from MRI scans [4]. The secondary goal of the study was to compare the accuracy of different segmentation tools, including that of a commercial semi-automatic muscle segmentation toolbox (Mimics 26.0, Materialise). The contributions of this study are: 1) A new mean shape (MS) based post-processing method is proposed, utilizing Statistical Shape Modelling (SSM) to fine-tune the segmentation output, taking into consideration the muscle anatomical shape; 2) A comparative study is conducted to investigate the effectiveness of post-processing approaches to the improvement of the DL automatic segmentation accuracy in terms of DSC, HD and ASSD.

## 2 Materials and methods

### 2.1 Participants

The data from two different cohorts of healthy post-menopausal women (PMW) were used in the study. The first cohort (PMW-1) includes 10 PMW T1-weighted magnetic resonance images with no muscle disease [19] and the second (PMW-2) was recruited from a previous observational study and consisted of 8 healthy PMW without muscle diseases [20]. The detailed parameters of both cohorts are given below (Table 1). Although the two cohorts were imaged in separate studies, the scans were conducted at the same hospital using two different 1.5T MRI scanners, following the same scanning protocol. All complete lower-limb MRI scans were performed using the 1.5T Siemens Magnetom Avanto or the 1.5T Siemens Magnetom Aera (Siemens AG, Erlangen, Germany). Four sequences were acquired to image the hip,

**Table 1. Statistical information.**

| Cohort | Age (year) | Weight (kg) | Height (cm) | BMI |
|---|---|---|---|---|
| PMW-1 | 69.0±6.7 | 66.9±7.7 | 159±3 | 26.5±3.4 |
| PMW-2 | 65.8±3.9 | 57.1±5.8 | 161±3 | 22.0±2.1 |

thigh, knee, and shank [21]. To reduce scanning time without compromising the detailed geometry of the joints, the joints were imaged at a higher resolution (pixel size of 1.05 mm × 1.05 mm and slice thickness of 3.00 mm) compared to the central portions of the long bones (pixel size of 1.15 mm × 1.15 mm and slice thickness of 5.00 mm). These sequences were then combined using MATLAB (R2006a) to create a continuous 3D image from the hip to the ankle. This was achieved by standardizing the resolution of each imaging sequence from the different sections through tri-linear interpolation (interp3, MATLAB R2006a) to 1.00 mm × 1.00 mm × 1.00 mm.

To generate the training and testing data for U-Net model, manual MRI segmentation was conducted. During manual segmentation operated by experts, the inter- and intra-operator reproducibility was found to differ from muscles to muscles (range CoV: 4.2%-22.8%), and only some of them were used to validate and calibrate the DL models [19]. In this study, 25 muscles from PMW-1 from knee to hip were used in model training period and only 16 muscles which with higher reproducibility were selected for further analysis: rectus femoris (RF), vastus intermedius (VI), vastus lateralis (VL), vastus medialis (VM), sartorius (SAT), semimembranosus (SMB), semitendinosus (SMT), gracilis (GRA), biceps femoris caput brevis (BCB), biceps femoris caput longum (BCL), adductor magnus (AM), adductor brevis (AB), adductor longus (AL), gluteus maximus (GM), iliacus (IL), and tensor fasciae latae (TFL).

The MR images of lower-limb muscles from PMW-1 were segmented by operators (two PhD students and one postdoctoral researcher) using the semi-automated muscle segmentation toolbox in Mimics 20.0 (Materialise, Leuven, Belgium), followed by some manual adjustments and post-processing [19]. The scans from PMW-2 were segmented by one operator (one PhD student) using the software Amira (version 2022.2; Thermofisher). All the operators have similar technical levels in terms of muscle segmentation and were trained on at least 15 datasets.

## 2.2 Experimental design and U-Net

U-Net [1] was used as the deep learning (DL) network during the experimental phase, the model layers setting detail is given in Table 2 and the flowchart of structure is shown in S1 Fig.

**Table 2. U-Net layers setting.**

| | | | |
|---|---|---|---|
| 1 | Im input 256x256x1 | 18 | Depth concat of 2 inputs |
| 2 | 64 3x3 convs, stride [1,1], BN + ReLU | 19 | 512 3x3 convs, stride [1,1], BN + ReLU |
| 3 | 64 3x3 convs, stride [1,1], BN + ReLU | 20 | 512 3x3 convs, stride [1,1], BN + ReLU |
| 4 | 2x2 max pool stride [2,2] | 21 | 256 2x2 transp convs stride [2,2], BN +ReLU |
| 5 | 128 3x3 convs, stride [1,1], BN + ReLU | 22 | Depth concat of 2 inputs |
| 6 | 128 3x3 convs, stride [1,1], BN + ReLU | 23 | 256 3x3 convs, stride [1,1], BN + ReLU |
| 7 | 2x2 max pool stride [2,2] | 24 | 256 3x3 convs, stride [1,1], BN + ReLU |
| 8 | 256 3x3 convs, stride [1,1], BN + ReLU | 25 | 128 2x2 transp convs stride [2,2], BN +ReLU |
| 9 | 256 3x3 convs, stride [1,1], BN + ReLU | 26 | Depth concat of 2 inputs |
| 10 | 2x2 max pool stride [2,2] | 27 | 128 3x3 convs, stride [1,1], BN + ReLU |
| 11 | 512 3x3 convs, stride [1,1], BN + ReLU | 28 | 128 3x3 convs, stride [1,1], BN + ReLU |
| 12 | 512 3x3 convs, stride [1,1], BN + ReLU | 29 | 64 2x2 transp convs stride [2,2], BN +ReLU |
| 13 | 2x2 max pool stride [2,2] | 30 | Depth concat of 2 inputs |
| 14 | 1024 3x3 convs, stride [1,1], BN + ReLU | 31 | 64 3x3 convs, stride [1,1], BN + ReLU |
| 15 | 1024 3x3 convs, stride [1,1], BN + ReLU | 32 | 64 3x3 convs, stride [1,1], BN + ReLU |
| 16 | 2x2 max pool stride [2,2] | 33 | 26 1x1 convs stride [1,1] |
| 17 | 512 2x2 transp convs stride [2,2], BN +ReLU | 34 | Softmax |

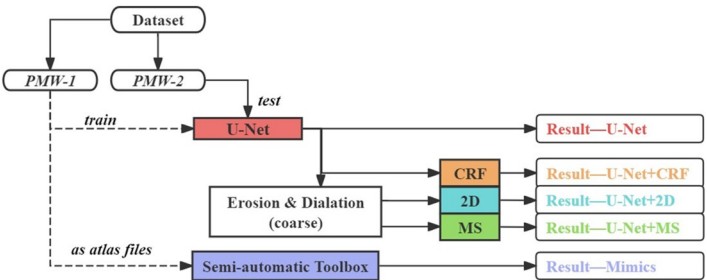

**Fig 1. Experimental design pipeline.**

During the training period, the dataset was randomly split into training and validation sets with a 9:1 ratio. The initial learning rate for the model was set to 0.001, and it decayed by 0.9 every 5 epochs, for a total of 100 epochs. The model was trained with the multi-class cross entropy loss function. Other hyper-parameters were tuned empirically for the optimal network performance, whilst ensuring the GPU memory and capacity were not exceeded. After the loss converged on the training set, the best-performing set of weights from each epoch on the validation set was retained. PMW-1 cohort was employed as the training set to train the U-Net, and then PMW-2 cohort was used as the test set to obtain preliminary results as original results (Fig 1). This entire process was repeated three times. The model's original prediction results were retained as a control group and used as input for post-processing. After obtaining the U-Net output results, rough 3D erosion and dilation operations were used in MATLAB to remove some obvious local errors. Subsequently, 2D post-processing (in section 2.5.2) [6] or the mean shape-based post-processing method (in section 2.3) proposed in this study was applied to obtain "Result—U-Net+2D and "Result—U-Net+MS" (Fig 1), respectively. As an alternative approach, the label map obtained from the last SoftMax layer of U-Net was used as input into the CRF post-processing method (in section 2.5.1), obtaining "Result—U-Net +CRF". As there were three trials in total, the muscle score of each metric for every subject was defined as the average of the three repetitions. In the semi-automatic segmentation toolbox (Mimics, 26.0, Materialise, Leuven, Belgium) (in section 2.4), PMW-1 cohort was used as the atlas file. Initially, the corresponding muscle areas in each lower limb MR image of PMW-2 were manually selected using thresholding, followed by simulation to obtain the results as "Result—Mimics" (Fig 1).

## 2.3 Proposed mean shape (MS) based post-processing

Considering that the same muscles in different individuals have similar 3D shapes and topological structures [14, 15], this study proposed a post-processing method based on SSM. This method aimed to further optimize the predicted segmentation results of the DL model by utilizing the MS obtained from SSM in combination with deformable registration using a nonlinear deformable image registration algorithm (ShIRT) [21, 22].

To establish homologous surface points in a dataset, surfaces were sampled to represent correspondence points. These points were encoded as a $3n$ vector set for shape representation and statistical analysis, with accuracy depending on point selection quality. Generally, corresponding points are selected manually as landmarks on the target surface, but the growing data volumes make manual selection more cumbersome and unsatisfactory [18]. Parametric methods have been used to address this issue, such as the one proposed in [23], which used a method based on fixed spherical parameterization mapping to select points. However, most

organs' shapes cannot be accurately parameterized, so these methods still have limitations. Cates [18] proposed a non-parametric analysis system——Shape Modelling with Entropy-Based Particle Systems (ShapeWorks correspondence optimization algorithm), which can optimize the distribution of a set of shape-similar correspondences and seeks a compact distribution of corresponding points in the shape space through an optimization algorithm to simplify the model. The basic idea of ShapeWorks is to construct a set of dynamic particles, each representing a set of correspondence points constrained on the surface of one shape. By balancing the negative entropy of particle distribution on each shape surface and the entropy of each shape's distribution in shape space, and using gradient descent, the positions of the correspondence can be optimized.

Assume having $N$ samples with $m$ correspondences on each surface which can be seen as a $3 \times 1$ vector in shape space, $x_k$ represents a set of particles from $k$th shape, $x_k \in R^{3m}$, $x_k = \{x_k^1, x_k^2, x_k^3, \ldots, x_k^{3m}, \}$, $k = \{1, 2, \ldots, N\}$ and let $Z = \{x_1, x_2, \ldots, x_N\}$, $Z$ is a random variable of a shape in the shape space after aligning all shapes to the same coordinate system. Then the method is to consider minimizing the energy function where $H$ is an estimation of differential entropy.

$$Q = H(Z) - \sum_N H(x_k) \tag{1}$$

To enhance the accuracy of the model, the first term in $Q$ minimization targets a compact distribution of individual's sample in the shape space. Meanwhile, the second term aims to increase the entropy of the correspondence distribution by achieving a more uniform distribution of points on each shape. The algorithm balances individual shape variation and overall shape similarity by minimizing both terms, allowing it to accurately model shape variations in the dataset.

In this study the non-parametric analysis system proposed by Cates was used, using the publicly available software tool ShapeWorks [23]. To use the software, $n$ different subject's shape models of the same muscle are provided as input. The software then generates a mean shape model, from which $n$-1 modes are extracted using principal component analysis (PCA). Multiple deformations are produced based on the average shape model with different modes.

The detail operation is given in Fig 2.

After producing the MS of individual muscle and the alignment operation in MATLAB, the ShIRT tool was used to do deformable image registration between the shape predicted by the model and the corresponding MS.

ShIRT is a non-linear deformable image registration algorithm that effectively tackles the significant anatomical variability commonly found in medical images [21, 22]. Its unique approach involves calculating a displacement function that maps every voxel in the reference image to its corresponding point in the target image. Through an iterative optimization process, the algorithm further refines the mapping matrix to minimize the cost function, which is determined by summing the squared differences between the intensity values of the two images. ShIRT calculates node displacements on an isotropic hexahedral grid overlaid onto fixed and moving images, with node-to-node distance Nodal Spacing (NS). Optimal nodal displacements are continuously smoothed during registration using a coefficient $\lambda$, to solve the registration problem efficiently. To generate a 3D displacement field, the algorithm uses trilinear interpolation between grid nodes. This field is then applied to transform the reference image. In a previous sensitivity analysis, the optimal value for the NS was found to be equal to 5 voxels (5 mm) for muscle segmentation tasks [21]. During hyper-parameter tuning, the Nodal Spacing (NS) was set to 5 mm and the smoothing coefficient $\lambda$ was automatically optimised by software [21].

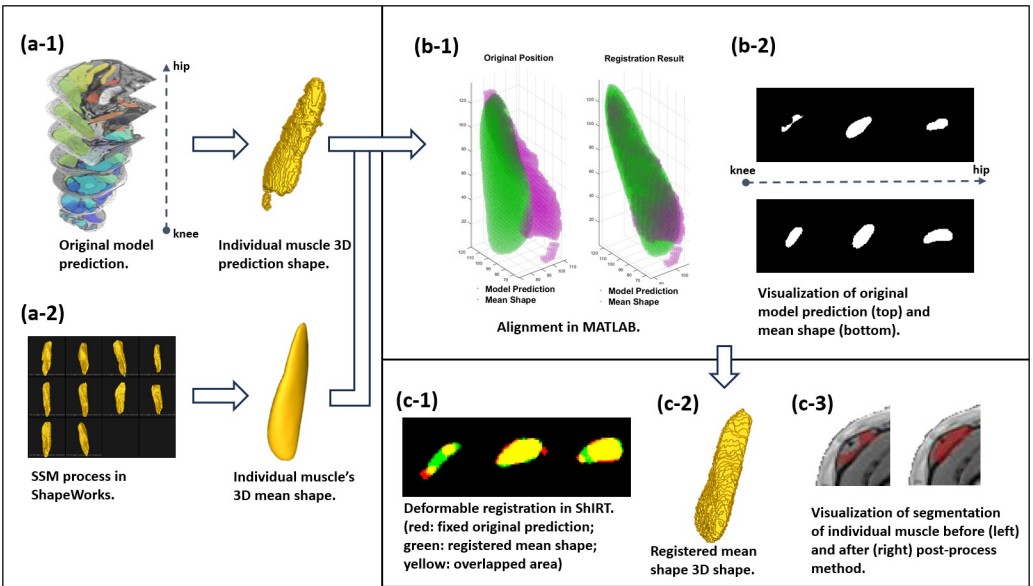

**Fig 2. Pipeline for mean shape based post-processing method.** Initially, the muscle segmentation results (original predictions) from the neural network were stacked, and subsequently, each muscle label was extracted to generate distinct 3D shapes (a-1). Using the ShapeWorks software, the mean shape for each muscle was generated (a-2) using gold standard manual segmentation from PMW-1. To reduce substantial spatial disparities between the original prediction and the mean shape, a rigid registration based on an iterative closest point algorithm (ICP) was performed in MATLAB (fixed shape: original prediction; moving shape: mean shape; function: pcregistericp (moving, fixed)) (b-1). Variations in muscle (e.g., TFL) slices at corresponding positions of both, from the knee to the hip, are depicted (b-2). In the final step, the two shapes underwent deformable registration in ShIRT, and the registered mean shape was retained as the post-processing segmentation results (c-1/2/3).

## 2.4 Semi-automatic muscle segmentation in Mimics for comparison

The Muscle Segmentation toolbox in Mimics (Mimics 26.0) enables to semi-automatically segment many muscles of the lower limb at the same time, rather than segmenting them one by one. Semi-automatic means that it is necessary to manually adjust the parameters to select the segmentation muscle area before running the software.

Before using the toolbox, it is necessary to create atlas files using already segmented samples, which serve as references for the software during segmentation. The atlases include MR scan images with mask (Fig 3(a)) corresponding to the areas to be segmented, as well as the mask of each segmented muscle (Fig 3(b)). Before running the program, it is also necessary to

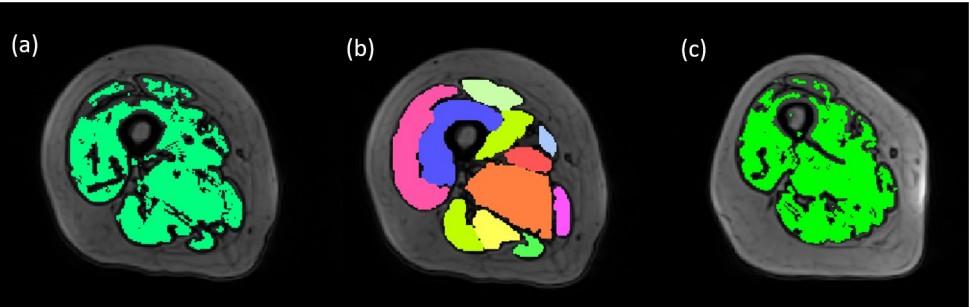

**Fig 3. The mask and segmentation visualization of atlas and new subjects in Mimics.**

manually add a mask (Fig 3(c)) on the original image and use thresholding and gradient thresholding to cover the muscle area with the mask as much as possible and eliminate the impact of background noise. Then, the MR images of new samples are input into the toolbox, and the segmentation results of each muscle are obtained after running the algorithms embedded in the software.

## 2.5 Existing post-processing methods for comparison

**2.5.1 Fully connected conditional random fields (CRFs).** In [13], a fully connected CRF method that utilized pairwise potentials among all pixels in the image was proposed to compensate for the restriction in complexity of inference and the limitation from unsupervised segmentation [24, 25]. The pairwise edge potentials were defined using a set of Gaussian kernels in an arbitrary feature space. This method was based on a mean field approximation to the CRF distribution which was optimized through message passing steps [13] to avoid the sudden increase in computation caused by image size. Fully connected conditional random fields conform to the Gibbs distribution [26], as shown in the following formula:

$$P(x = X|I) = \frac{1}{Z(I)} e^{-E(X|I)} \tag{2}$$

where $x$ is the observed value, $I$ is the colour information of pixel, $E(X|I)$ is the energy function which consists of a unary potential function and a binary potential function, as shown in the following formula:

$$E(x|I) = \sum_i \psi_u(x_i) + \sum_{i,j} \psi_p\left(x_i, y_j\right) \tag{3}$$

where $x_i, y_j$ are the labels assigned to pixel $i$ and $j$, respectively. The unary potential function is used to measure the probability of the observed pixel to different classes which is from the output feature maps of CNNs. A binary potential function measures the probability $\psi_p(x_i, y_j)$ of two events occurring at the same time where if the pixel values are very close, then the probability that the two pixels belong to the same category should be relatively high; on the contrary, if the colour difference is relatively large, then the probability of these two pixels belong to different category should be relatively high. This energy term is designed to split the segmentation results as far as possible from the edges of the image or between different targets.

$$\psi_p\left(x_i, y_j\right) = u\left(x_i, y_j\right) \sum \omega^m K_G^m\left(f_i, f_j\right) \tag{4}$$

$u(x_i, y_j)$ is the label compatibility item which restricts the conditions of conduction between pixels, and only under the same label conditions can energy be transmitted to each other. $\omega^m$ is the weight parameter. $K_G^m(f_i, f_j)$ is Gaussian kernel including appearance kernel and smoothness kernel in terms of colour vector $I_i$ and $I_j$ and positions $p_i$ and $p_j$, and defined as below:

$$K_G^m\left(f_i, f_j\right) = W^{(1)} e^{-\frac{|p_i - p_j|^2}{2\theta_\alpha^2} - \frac{|I_i - I_j|^2}{2\theta_\beta^2}} + W^{(2)} e^{-\frac{|p_i - p_j|^2}{2\theta_\gamma^2}} \tag{5}$$

$W^{(1)}$ and $W^{(2)}$ are the weight of appearance kernel and smoothness kernel, respectively. The concept behind the appearance kernel stems from the observation that adjacent pixels sharing similar colours are probable candidates for belonging to the same class. The weights of nearness and similarity are controlled by the parameters $\theta_\alpha$ and $\theta_\beta$. The smoothness kernel removes small, isolated regions and is controlled by $\theta_\gamma$. The detailed derivation of CRFs can be found in

**Table 3. Parameters used in the 2D post-processing method.**

| Operation | MATLAB function | Parameters |
|---|---|---|
| Erode | imerode(label, SE) | $r = 3$ |
| | SE = strel('disk',r) | |
| Bwlabel | bwlabel(label,conn) | conn = 8 |
| Dilate | Imdilate(label,SE) | $r = 3$ |
| | SE = strel('disk',r) | |
| Fill holes | imfill(label,'holes') | |
| Smoothness | kernel = ones(w)/w^2 | $w = 5$ |
| | image = conv2(single(label),kernel,'same') | threshold = 0.5 |
| | image = image>threshold | |

[13]. In Python, the CRF method can be implemented using the 'pydensecrf' library provided where $W^{(1)}$, $\theta_\alpha$ and $\theta_\beta$ was set to 10, $W^{(2)}$ and $\theta_\gamma$ was set to 1 during learning in [13].

**2.5.2 2D post-processing method.** In this algorithm, an erosion operator (function: imerode, in MATLAB) was first used to remove small noises from the main cohort, and then the 2D connected components were found (using function: bwlabel). Assuming the target area is a single volume and with a smooth surface, the biggest area was kept and the empty holes inside it were filled (function: imfill, dilate) and a 2D convolution with a uniform square filter was used to achieve the smoothness (function: conv2D). The operations were based on each predicted 2D slice and tuning details are given below (Table 3) [6]. Unlike the single-class segmentation of the liver, the multi-class muscle segmentation conducted in this study entails that each muscle is separately extracted for post-processing and then ultimately pieced together.

## 2.6 Evaluation metrics and statistics

Four metrics were used in this study: Dice Similarity Coefficient (DSC), Relative Volume Error (RVE), Hausdorff Distance (HD), and Average Symmetric Surface Distance (ASSD) [2, 21, 27]. These metrics can be used to comprehensively evaluate the accuracy of segmentation results (DSC and RVE) and assess the impact of incorrect segmentation region and local error (HD and ASSD).

The DSC shows the similarity between reference shapes (R) and predicted segmentation shapes or post-processed shapes (P) and was calculated by using Eq (6)

$$DSC = \frac{2|R \cap P|}{|R| + |P|} \tag{6}$$

where $R$ is the set of voxels in the reference labels and $P$ is from prediction or post-processed results.

The RVE was used to assess the similarity between the ground truth $V_{ref}$ and the segmentation results $V_{pred}$ in terms of the total volume of each muscle. RVE is defined as

$$RVE = \frac{V_{pred} - V_{ref}}{V_{ref}} \times 100\% \tag{7}$$

where $V_{ref}$ and $V_{pred}$ are the volumes of reference and prediction cohort respectively.

The HD is a metric that measures the maximum distance of external surfaces between the reference and predicted shapes. It evaluated the effect of the extent of local errors in the

model's prediction and was calculated as

$$HD(R, P) = max\{d(R, p), d(r, P)\} \tag{8}$$

where $r$ is the voxel on the surface of $R$ and $p$ is the voxel on the surface of $P$, $d$ is a function to find the minimum distance between $r$ or $p$ and the nearest point within the cohort $P$ or $R$.

The ASSD measures the mean distance of surface voxels between the predicted or post-processed label boundaries and the reference label boundaries. The ASSD is defined as

$$ASSD = \frac{\sum_{p \in B_P} d(p, B_R) + \sum_{r \in B_R} d(r, B_P)}{B_R + B_P} \tag{9}$$

where $B_R$ and $B_P$ are the boundaries of reference labels and predicted or post-processed labels, respectively. $d(x, Y)$ is the Euclidean distance between voxel $x$ and surface $Y$. A smaller ASSD score, closer to 0, indicates that the segmentation results are closer to the gold standard.

The 8 testing subjects were also evaluated separately using these 4 metrics to observe the bias effect caused by individual cohort to the main performance.

Wilcoxon signed rank test [4] was used to statistically test differences for each metrics between the different methods because when compare the difference between the paired values, they were not normally distributed. A significance level $a = 0.05$ was considered.

## 3 Results

### 3.1 Performance comparison under different methods

In total, 128 labels representing one individual lower-limb muscle (8 subjects in PMW-2 with 16 muscle labels each) were evaluated and used to compare the performance of each post-processing method or semi-automatic segmentation tool with original prediction results from the trained U-Net.

It can be observed that original results from U-Net, and results from U-Net+2D, and U-Net +MS exhibit jointly optimal results in terms of DSC (Fig 4), averaging around 0.826 (0.825–0.827), representing an improvement of 1.7% (p<0.01) and 22.7% (p<0.01) over U-Net+CRF (0.812) and Mimics (0.673), respectively.

The U-Net+2D showed the best performance in RVE, with its absolute values being closest to zero, indicating a high degree of alignment with the gold standard in terms of volume prediction. It outperformed the other methods with a percentage difference between the absolute values of the RVE for one method and U-Net+2D from 2.4% to 9.0% (p<0.05). U-Net+MS shows the worst RVE performance (0.109 ±0.162).

U-Net+MS exhibited the best performance in terms of HD (mean value: 20.6mm ±11.7mm), better than U-Net (78% improvement, p<0.01), Mimics (53.8% improvement, p<0.01), U-Net+2D (39.8% improvement, p<0.01) and U-Net+CRF (43.3% improvement, p<0.01). Similarly, the U-Net+MS was associated with the lowest ASSD (mean value: 2.1mm ±0.9mm), better than U-Net (53.3% improvement, p<0.01), Mimics (65.6% improvement, p<0.01), U-Net+2D (19.2% improvement, p<0.01) and U-Net+CRF (47.6% improvement, p<0.01).

The results of the different methods for the 8 testing individual subjects confirmed similar variability and trends across them (S2 Fig). When results are analysed for individual subjects (Supplementary materials) similar trends were found as when results were averaged across the cohort for each method (Fig 4). It is shown that, there is minimal individual bias existed causing impact on the result in this experiment.

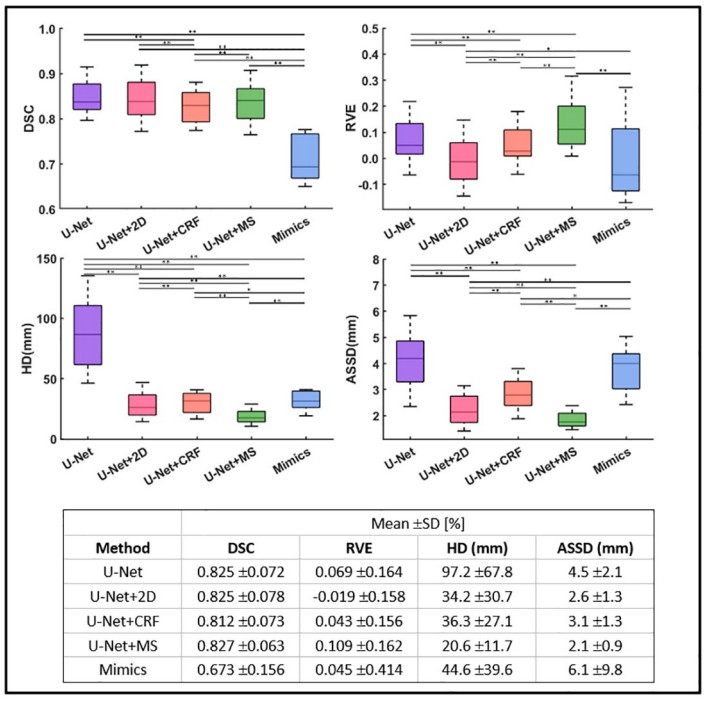

**Fig 4. Performance comparison of each method.** Box plots for the DSC (top left), RVE (top right), HD (bottom left), and ASSD (bottom right) for each method (16 medium values of each muscle calculated from 128 labels) are shown (* indicates p<0.05, ** indicates p<0.01). Lower scores indicate better performance in HD and ASSD. U-Net denotes original prediction from U-Net; U-Net+2D denotes 2D post-processing; U-Net+CRF denotes fully connected conditional random fields; U-Net+MS denotes mean shape based post-process method; Mimics denotes the results from Mimics semi-automatic muscle segmentation toolbox.

When the labels of the 16 target muscles were pooled into one label, DSC, RVE, and ASSD were used to evaluate the recognition performance of each method on the overall target muscle region (Fig 5). Because the whole muscle cohort is larger and longer than the individual muscle, the comparison of HD is meaningless and therefore removed.

The U-Net presents the highest DSC value of 0.911 ± 0.009, indicating the best performance in terms of the accuracy of segmentation compared to the ground truth. Mimics has the lowest DSC value (0.862 ± 0.038), showing a significant reduction in segmentation accuracy by 5.4% (p<0.01) compared to the U-Net. The other methods (U-Net+2D, U-Net+CRF, U-Net+MS) show slightly lower DSC values than U-Net, indicating minor deviations in segmentation accuracy with U-Net+2D at 0.901 ± 0.014 (p<0.01), U-Net+CRF at 0.905 ± 0.010 (p<0.01), and U-Net+MS at 0.902 ± 0.006 (p<0.05).

2D post-processing method demonstrates the best performance in RVE, with its value being the closest to zero (-0.007 ± 0.057), indicating the highest accuracy in volume prediction relative to the gold standard. The U-Net+MS method still shows the worst performance in RVE (0.096 ± 0.057), suggesting a deviation from the gold standard in volume prediction compared to other methods.

The U-Net+MS approach exhibits the best performance in terms of ASSD, with the lowest mean value (2.0 mm ± 0.2 mm), indicating the highest precision in the segmentation boundary compared to the gold standard. In contrast, U-Net shows the highest ASSD (8.2 mm ± 1.6 mm). The U-Net+2D, U-Net+CRF, and Mimics methods show intermediate performances

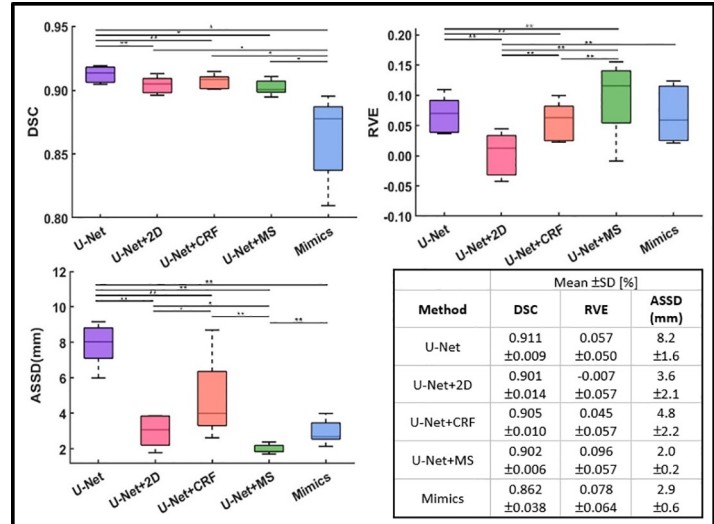

**Fig 5. Performance comparison of each method after pooling all labels into one.** Box plots for the DSC (top left), RVE (top right), and ASSD (bottom left) for each method (8 values from 8 subject under each metric) are shown (* indicates p<0.05, ** indicates p<0.01).

with ASSD values of 3.6 mm ± 2.1 mm, 4.8 mm ± 2.2 mm, and 2.9 mm ± 0.6 mm, respectively, reflecting varied levels of boundary precision in segmentation.

## 3.2 Relationship between muscle volume and segmentation accuracy

As muscle volume increases, the accuracy of segmentation under each method also shows an increase. For muscles with large volume differences between samples, such as VI, or for smaller and irregularly shaped muscles like AB and BCB, the segmentation precision is low.

## 3.3 Visualization

Compared to the gold standard (column 0, Fig 6), the U-Net segmentation results from each method primarily exhibited two issues: blank regions and mis-segmentation. Blank regions are defined as areas that should belong to a certain muscle class but have not been assigned a value, resulting in empty spaces (such as indicated in red arrow in a-1, b-1), mis-segmentation is defined as areas that should be classified as muscle A but are identified as muscle B or muscle C volumes (such as indicated in yellow arrow in a-2, b-2).

In the original predictions of the U-Net, as illustrated in Fig 6a-1, there were substantial blank areas in the results related to VM. After 2D post-processing method (Fig 6a-2) the issue persisted, and, conversely, there were less regions well identified. However, results obtained through U-Net+CRF (Fig 6a-3), and U-Net+MS (Fig 6a-4) showed improvement, with some small areas correctly filled.

The original results also exhibit problems of mis-segmentation in the SAT and SMB muscles. In Fig 6a-1, the SAT prediction was entirely mis-segmented, and the SMB predicted area included a substantial region of incorrectly segmented muscles. In U-Net+2D (Fig 6a-2) and U-Net+CRF (Fig 6a-3) results, errors in the SAT segment were removed, leaving behind blank areas. However, in the SMB area, incorrect prediction areas were not fully removed, and correct predictions were mistakenly removed in some cases. MS effectively addressed these issues (Fig 6a-4), where both SMB and SAT muscle regions were relatively well segmented. The same

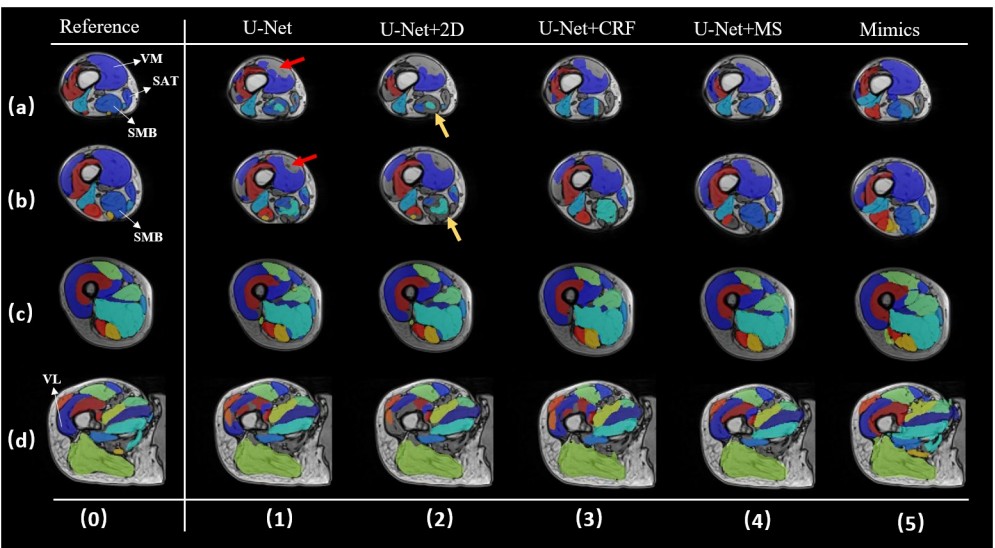

**Fig 6. Segmentation performance visualization of each method.** The figure exhibits the segmentation performance of each method on one subject from a location close to knee to hip, rows a-d. Column 0 denotes the golden standard, and columns 1–5 denote different segmentation results.

issue was well-reflected in the SMB area (Fig 6b). After U-Net+2D or U-Net+CRF, the SMB area worsened, whereas the U-Net+MS method reduced errors.

The U-Net prediction result (Fig 6d-1) showed errors in the VL area which incorrectly divided the properly segmented VL area into two halves (yellow arrows). Consequently, in the U-Net+2D, one-half of the correct VL predicted area was mistakenly removed. The U-Net +MS approach (Fig 6d-4) was found to be superior compared to the U-Net+2D (Fig 6d-2) and U-Net+CRF (Fig 6d-3) approaches.

When the labels of 16 target muscles are treated as one label, the U-Net+MS method can correct the empty spaces in the U-Net results (as indicated by the red arrow in Fig 7c-1 and 7d-1). Except for Mimics, the other methods do not present a large difference in this visualization case.

Fig 8 provides a direct intuition of how post-processing step reduces HD and weakens the side effects from local errors. The segmentation after post-processing step is more similar to GT in geometry, especially at both ends of each muscle, and post-processing step can improve obvious unreasonable areas (e.g. in Fig 8g) leading to lower HD score with better performance.

## 4 Discussion

This study aimed to propose a novel muscle segmentation post-processing method based on mean shape from a statistical shape model to improve the accuracy of the deep learning segmentation results. Overall, the new method, which uses the mean shape and deformable registration, has been found to be significantly more accurate than the original prediction from the U-Net, the post-processed results from selected 2D post-processing and CRF methods, and the direct segmented results from muscle semi-automatic tool from Mimics.

U-Net segmentation followed by MS post-processing demonstrated the best performance in terms of HD (20.6mm) and ASSD (2.1mm) and had similar DSC score (0.827) compared to the original (U-Net, no postprocessing) results (0.825) and U-Net segmentation followed by 2D post-processing method (0.825). However, U-Net+MS is also associated with the highest

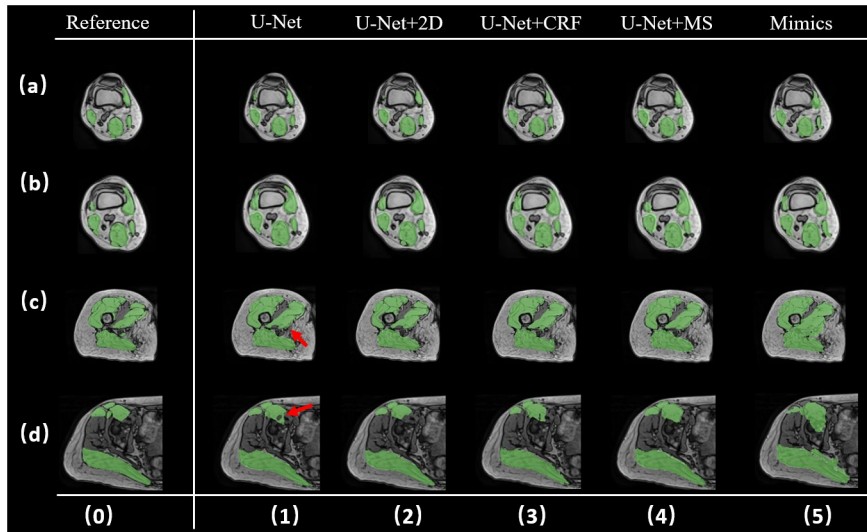

**Fig 7. Segmentation performance visualization of each method after pooling all labels into one.** The figure exhibits the segmentation performance of each method on one testing subject from a location close to knee to hip, rows a-d. Column 0 denotes the golden standard, and columns 1–5 denote different segmentation results.

RVE (0.109) and a systematic overestimation of the muscle volumes, as also confirmed by the high RVE for pooled labels (Fig 5). This indicates that, while using the MS method on U-Net prediction results can provide better corrections for some local mis-segmentation points and large outlying points, this approach is less accurate than others when the correct estimation of the volume of the muscles is required.

Compared to U-Net+MS results, the results from Mimics have lower RVE but on average lower DSC scores. Fig 9 shows the Mimics segmentation results (green) of two muscles (Vastus Intermedius and Sartorius) compared with the ground truths (purple), where the RVE scores of both muscles were good, 0.1% and 0.15%, but the DSC scores were lower than 0.70, 0.571 and 0.684. The lower RVE and the lower DSC normally indicates that there are large mis-identified volumes (non-overlap volumes) but that the combination of overestimated volume in one region and underestimated volume in another region lead to similar overall volume, as shown in Fig 9. This issue confirms that reporting only the RVE in image segmentation studies is not enough and could lead to misinterpretation of the results.

The proposed MS method retains the mean shape after deformable registration, making it superior in handling local errors compared to other methods in the control group, as evidenced by improvements in HD and ASSD. In 2D post-processing method, a 4% performance improvement was achieved in liver segmentation. However, compared to the liver, the individual muscle segmented in our study is more complex in size, shape, texture, and surface variations, leading the segmentation more challenging. In fact, when all muscles labels are pooled together, better metrics (DSC: 0.91) can be observed (Fig 5). It is worth noting that the volume-related accuracy (RVE: -0.019) is highest for the U-Net+2D approach. The comparison between the predicted muscles from the U-Net approach alone or followed by 2D post-processing approach (Fig 6d-1 and 6d-2) suggests that for certain type of muscles involving more than two independent regions, the 2D post-processing method will result in poorer outcomes. This limitation of the 2D post-processing method is due because it only retains the larger region of the muscle. The CRF algorithm itself relies more on the predicted label map relationships between target pixels and surrounding pixels [13, 26, 28], sometimes overlooking

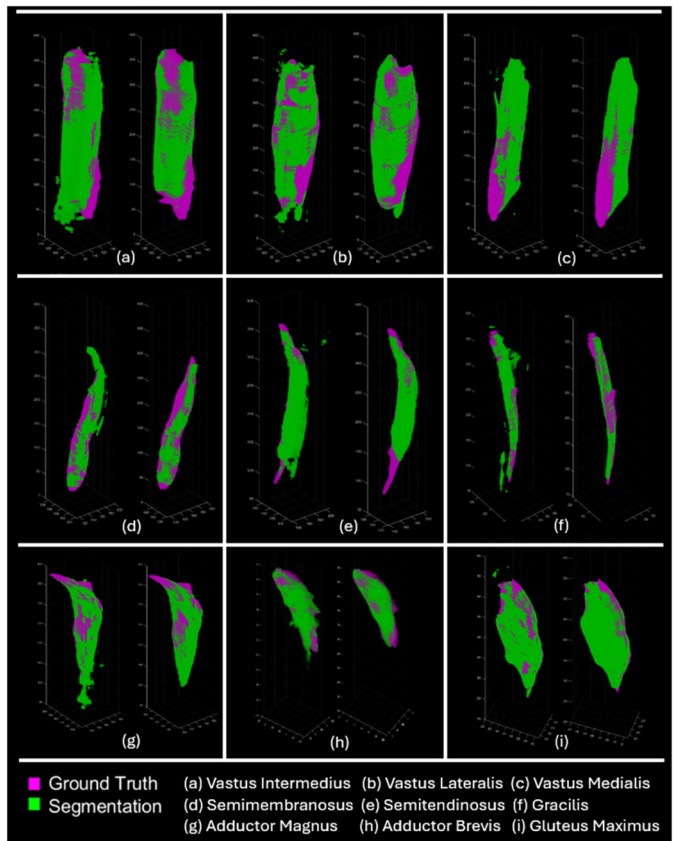

**Fig 8. Comparison between ground truth and segmentation results with/without post-processing step.** GT (in purple) and segmented results (in green) are overlapped and visualized by converting them into point clouds in MATLAB. Parts a-i are the comparison of VI, VL, VM, SMB, SMT, GRA, AM, AB and GM from one testing subject, respectively. In each part, the left side is the comparison of U-Net results with GT, and the right side is the comparison of U-Net+MS results with GT.

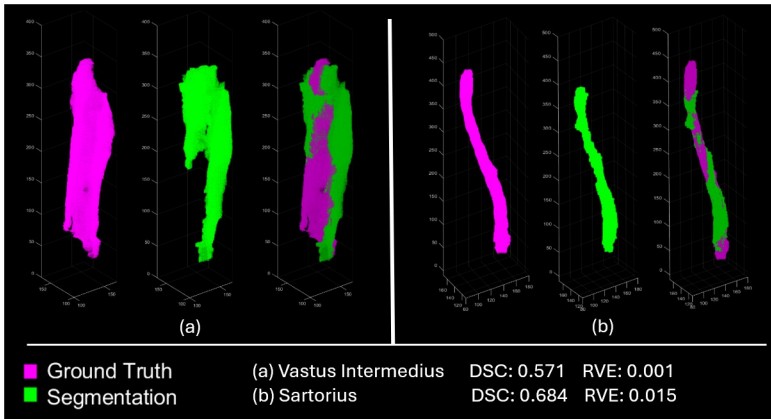

**Fig 9. Comparison between ground truth and segmentation results from Mimics.** In each part, the ground truth (left), segmentation result (middle) and the overlapping section of both (right) are shown. Parts a and b represent the results of muscle vastus intermedius and sartorius of same testing subject.

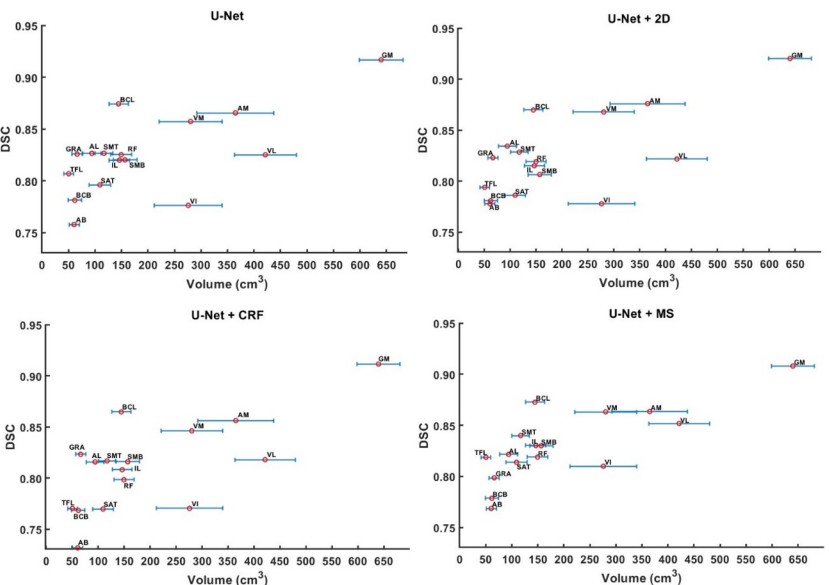

**Fig 10. Relationship between muscle volume and DSC score under 4 methods (U-Net (top left), U-Net+2D (top right), U-Net+CRF (bottom left) and U-Net+MS (bottom right)) of test subjects.** The muscles were arranged in ascending order according to the mean muscle volume of PMW-2 cohort. The error bar of each point denotes the standard deviation of muscle volume among 8 test subjects.

anatomical significance and relying solely on mathematical inference. This could lead to obvious errors, as seen in the SMB area (Fig 6A-3). In contrast to the literature [13], which used CRF based post-processing method for the segmentation of natural objects such as birds and trees, where there is more contrast between the object to segment and the background, the differences between adjacent muscles, or between muscles and fat tissues, are sometimes not as clear in the MRIs and will affect the performance of CRF inference. In fact, CRF only achieved a DSC score of 0.812, which is worse than the untreated results of the DL model. As muscle volume increases (Fig 10), the DSC under each methods also increases, consistent with the results reported in [4]. For example, the DSC for the Gluteus Maximus (GM) reaches values above 90%. Nevertheless, for smaller muscles, with volume below 200 $cm^3$, a wide range of DSC scare was found, from 73% to 88%, highlighting the heterogeneous performance of the model. It is therefore very important to choose the right model for different applications. Based on the DSC scores obtained in this experiment, a recommendation table (Table 4) is given.

In Table 4, the best method which shows the best performance (DSC/HD/ASSD) under individual muscle among 4 methods (U-Net, U-Net+2D, U-Net+CRF and U-Net+MS) is recommended as the chosen method with the corresponding value reported on the right.

Different postprocessing approaches were found to be best suited to segment the 16 lower limb muscles considered in this study, depending on the muscle and the considered metric (Table 4). Therefore, depending on the different applications, the user can choose which postprocessing method to use in order to obtain the best results. It can be shown that the post-processing method has improved the results of most muscle segmentation where U-Net+MS was more widely recommended, especially for the local analyses.

Compared to the prediction results of U-Net+MS and U-Net, the results obtained from the automated part of the semi-automatic muscle segmentation toolbox provided by Mimics are lower than average level. It only approaches the gold standard more closely in RVE (-0.019)

**Table 4. Recommended methods based on different target muscles.**

| Muscle | Chosen method / DSC | Chosen method / HD(mm) | Chosen method / ASSD(mm) |
|---|---|---|---|
| Rectus Femoris | U-Net / 0.825 | U-Net + MS / 19.9 | U-Net + MS / 2.3 |
| Vastus Intermedius | U-Net + MS / 0.810 | U-Net + MS / 22.9 | U-Net + MS / 2.5 |
| Vastus Lateralis | U-Net + MS / 0.852 | U-Net + MS / 28.6 | U-Net + MS / 2.2 |
| Vastus Medialis | U-Net + 2D / 0.868 | U-Net + MS / 18.6 | U-Net + MS / 2.1 |
| Sartorius | U-Net + MS / 0.814 | U-Net + MS / 23.0 | U-Net + MS / 1.7 |
| Semimembranosus | U-Net + MS / 0.830 | U-Net + MS / 25.7 | U-Net + MS / 2.7 |
| Semitendinosus | U-Net + MS / 0.840 | U-Net + MS / 24.4 | U-Net + MS / 2.0 |
| Gracilis | U-Net / 0.826 | U-Net + MS / 15.2 | U-Net + MS / 1.5 |
| Biceps Femoris Caput Brevis | U-Net / 0.781 | U-Net + MS / 28.2 | U-Net + MS / 2.5 |
| Biceps Femoris Caput Longum | U-Net / 0.874 | U-Net + MS / 25.7 | U-Net + MS / 1.8 |
| Adductor Magnus | U-Net + 2D / 0.876 | U-Net + MS / 25.9 | U-Net + MS / 2.4 |
| Adductor Brevis | U-Net + 2D / 0.778 | U-Net + MS / 14.1 | U-Net + MS / 2.2 |
| Adductor Longus | U-Net + 2D / 0.834 | U-Net + MS / 15.3 | U-Net + MS / 2.0 |
| Gluteus Maximus | U-Net + 2D / 0.920 | U-Net + MS / 15.8 | U-Net + MS / 1.9 |
| Iliacus | U-Net + MS / 0.830 | U-Net + MS / 13.0 | U-Net + MS / 1.8 |
| Tensor Fasciae Latae | U-Net + MS / 0.819 | U-Net + MS / 13.4 | U-Net + MS / 1.7 |

and outperforms U-Net in HD (44.6mm). During the segmentation process, the software combines manually selected muscle regions using thresholding (as shown in Fig 3(a) and 3(c)) with pre-prepared atlas files for segmentation. This operation could be influenced by the differences between atlas files and the new input samples (i.e. increasing the number of files in the atlas is likely to improve the outcomes). The atlas files in this study were derived from PMW-1 cohort, while input samples were from PMW-2, with general consistency in age, volume ($p > 0.05$, S1 Table), etc. Therefore, the operation of Mimics in this study could be affected by individual differences between the two cohorts more than for the other methods. However, in the process of operation, the results automatically generated by Mimics are generally expected to be corrected manually by the operator, leading to better yet less efficient results.

As a crucial prerequisite for subsequent analysis and diagnosis, muscle segmentation plays a pivotal role in texture analysis (TA) [29] and quantitative assessment [19] of related parameters. In TA, any image features encompassed in the segmentation results can significantly impact conclusions. Therefore, the segmentation process should prioritize the removal of unnecessary local errors and noise-induced influences. Methods demonstrating superior performance in terms of HD and ASSD are better suited for TA. In quantitative analyses of muscles, such as mass and maximal isometric force [19], paying extra attention to accurate volume measurement is crucial throughout the segmentation process. Hence, the RVE metric assumes greater importance in the evaluation of segmentation results. Therefore, considering the four considered evaluation metrics, MS is more suitable for TA, while the 2D post-processing method is more applicable for quantitative analysis.

This study is affected by some limitations. Firstly, it is limited by the number of the datasets used (10 subjects for training and 8 subjects for testing). The number of unique images from each individual collected during MRI acquisitions was from 435 to 550 slices, leading to a total number of training slices of approximately 5000. However, considering that only 10 subjects were used for training, the heterogeneity of features in the muscles to be segmented was probably limited. As a result, when training DL models, the features learned by convolutional operation and the shape features on the average shape geometry in the MS method generated based

on the training set is probably incomplete. Although features including all samples in a specific range are not displayed, with the increase of data and the richness of subject types, the loss caused by incomplete feature learning can be reduced to a certain extent. In future work, more subjects from different cohorts will be collected to increase the range of muscle volumes and have larger distributions of muscle shapes. Larger datasets are likely also to increase the DL model performance as previously found in Rohm et al [30], the DSC score increase around 2% after adding augmentation data in Lin et al. [4]. In addition, the performance of the post-processing methods still depends on the quality of the results produced by previous methods such as DL model and automatic segmentation tools. The performance of the model will affect the quality of post-processing results, the better model which proved achieved better segmentation than U-Net, e.g. Attention-Feature-Fusion-UNet (AFFU) which is a modified U-Net added with attention mechanisms [4] achieved 2.0% gain in DSC compared to U-Net, and Spatial Channel UNet (SC-UNet) which considered the knowledge of where the image slice was acquired from within the lower limbs (along the axial direction) [5], can be used and linked with MS method in further study to observe is there any significant improvement.

## 5 Conclusion

This study has shown that using mean shape based post-processing method after an automatic segmentation with U-Net improves most metrics for segmenting individual muscles of the lower limb from MR images compared to the U-Net alone. In particular, this approach reduced the local segmentation errors compared to segmentations with U-Net alone or followed by post-processing methods based on 2D analyses or Conditional Random Field.

## Supporting information

**S1 Fig. U-Net structure.**
(TIFF)

**S2 Fig. Performance comparison of each method on individual testing subjects from PMW-2.** Subplots (s1-s8) represent the performance of four metrics (DSC/RVE/HD/ASSD) for each subject (8 in total) under different methods (each boxplot includes 16 individual muscles' score). The median number of each boxplot is connected by dotted red lines to see if the overall trend is the same.
(PDF)

**S1 Table. Muscle average volume comparison between PMW-1 and PMW-2.** The volume of each muscle in the table is achieved by averaging the corresponding muscle volumes across all subjects. There is no statistically significant difference in muscle volume between the two cohort.
(DOCX)

## Acknowledgments

We extend our sincere gratitude to Materialise UK for generously providing us with access to the Mimics software suite (Mimics & 3-matic). We appreciate the invaluable support and assistance provided by Materialise team (Materialise NV), which significantly contributed to the success of this study.

## Author Contributions

**Conceptualization:** Zhicheng Lin, Lingzhong Guo.

**Data curation:** Zhicheng Lin.

**Funding acquisition:** Enrico Dall'Ara.

**Methodology:** Zhicheng Lin.

**Project administration:** Enrico Dall'Ara, Lingzhong Guo.

**Software:** Zhicheng Lin, Enrico Dall'Ara.

**Supervision:** Enrico Dall'Ara, Lingzhong Guo.

**Validation:** Zhicheng Lin.

**Visualization:** Zhicheng Lin.

**Writing – original draft:** Zhicheng Lin.

**Writing – review & editing:** Enrico Dall'Ara, Lingzhong Guo.

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
