## [Decision Letter · Decision Letter 0]

11 Jun 2024

PONE-D-24-19418A novel mean shape based post-processing method for enhancing deep learning lower-limb muscle segmentation accuracyPLOS ONE

Dear Dr.  Guo,

Thank you for submitting your manuscript to PLOS ONE. After careful consideration, we feel that it has merit but does not fully meet PLOS ONE’s publication criteria as it currently stands. Therefore, we invite you to submit a revised version of the manuscript that addresses the points raised during the review process.

We look forward to receiving your revised manuscript.

Kind regards,

Fei Yan

Academic Editor

PLOS ONE

Journal Requirements:

   "The study was funded by Engineering and Physical Sciences Research Council (EPSRC, Frontier Multisim Grant, EP/K03877X/1, and EP/S032940/1), and the Medical Research Council (MRC) and Versus Arthritis (Medical Research Council Versus Arthritis Centre for Integrated Research into Musculoskeletal Ageing, CIMA, (MR/R502182/1). "

6. For studies involving third-party data, we encourage authors to share any data specific to their analyses that they can legally distribute. PLOS recognizes, however, that authors may be using third-party data they do not have the rights to share. When third-party data cannot be publicly shared, authors must provide all information necessary for interested researchers to apply to gain access to the data. (https://journals.plos.org/plosone/s/data-availability#loc-acceptable-data-access-restrictions) 

7. PLOS requires an ORCID iD for the corresponding author in Editorial Manager on papers submitted after December 6th, 2016. Please ensure that you have an ORCID iD and that it is validated in Editorial Manager. To do this, go to ‘Update my Information’ (in the upper left-hand corner of the main menu), and click on the Fetch/Validate link next to the ORCID field. This will take you to the ORCID site and allow you to create a new iD or authenticate a pre-existing iD in Editorial Manager. Please see the following video for instructions on linking an ORCID iD to your Editorial Manager account: https://www.youtube.com/watch?v=_xcclfuvtxQ

Reviewers' comments:

Reviewer's Responses to Questions

**Comments to the Author**

1. Is the manuscript technically sound, and do the data support the conclusions?

Reviewer #1: Yes

Reviewer #2: Yes

Reviewer #3: Yes

2. Has the statistical analysis been performed appropriately and rigorously? 

Reviewer #1: Yes

Reviewer #2: Yes

Reviewer #3: I Don't Know

3. Have the authors made all data underlying the findings in their manuscript fully available?

Reviewer #1: Yes

Reviewer #2: Yes

Reviewer #3: Yes

4. Is the manuscript presented in an intelligible fashion and written in standard English?

Reviewer #1: Yes

Reviewer #2: Yes

Reviewer #3: Yes

5. Review Comments to the Author

Reviewer #1: This study introduces a new post-processing method that compensates for the shortcomings of UNet-based segmentation methods based on Magnetic Resonance Imaging (MRI) scans and verifies its effectiveness in practice. The proposed method takes into consideration the muscle anatomical shape, enabling more effective and accurate segmentation of muscles. I believe that it is useful as it can be applied to other organ segmentation such as bone, lung etc. By comparing with existing post-processing approaches as well as a commercial segmentation software, it has demonstrated the good performance of the proposed post-processing approach. The paper is very well organised and written. Comments are:

1. The contribution of the study should be highlighted.

2. Obviously, the proposed post-processing approach can be apply in any segmented images. There are many different types of deep learning networks, can you explain why UNet was chosen as the deep learning model for this study?

3. In Participants, it looks like you used two different MRI scanners, were there any differences in imaging conditions between the two cohorts?

4. For the operators for manual segmentation, were they the same people or different people with similar technical level?

5. For the training, one cohort of data (PMW-1) was used as training data while the second cohort was used for test. Is there any population bias in the cohort which may affect your result?

6. In the proposed method, mean shape of the cohort (postmenopausal women) was used. What is your view for the method to apply to different population, say young people?

Reviewer #2: The manuscript presents a new post-processing method for the automatic segmentation of muscles in the upper leg. I find the approach to use a mean shape quite interesting. Overall the manuscript is well-written and has scientific merit. I do have some comments that I hope the authors can help clarify. Comments are (largely) listed in order of appearance in the manuscript:

General:

When data is pooled, is this done after the predictions? Or is a separate U-Net trained for the pooled data. If only 1 label is available for training the U-Net I imagine that you could achieve much better results compared to only pooling in post-processing.

Introduction:

Just reading the introduction, the goals of the study slightly confuse me. In L91 you mention that a new SSM-based post-processing method is proposed in this study. However, in the subsequent paragraph, you say that the goal of the study is to assess the effect of existing post-processing methods, with no mention of the proposed novel method. From the rest of the paper it is clear that this SSM-based method is an essential part of the goal of this study.

Methods:

More information regarding the MR images should be presented (pixel size, slice thickness, slice increment, FOV, any filtering applied)

L112-L124: it is unclear to me if the segmentations of PMW-1 were performed as part of this study, or in another study (reference 19). If I understand correctly, the scans for PMW-2 were segmented for this specific study. Were exactly the same muscles extracted? How was the segmentation performed? Was this also performed semi-automatically and performed by multiple operators and then compared? I believe that these segmentations are considered the golden standard mentioned in the results. I suggest mentioning this explicitly for clarity.

I suggest trying to put table 2 into a flowchart figure as is often done with U-Net networks.

L131: I'm a bit confused regarding how the U-Net was trained. Why was the U-Net trained and tested 3 times? Was it retrained for each method of post-processing? If so, wouldn't it make more sense to perform the post-processing on one trained model? Was there no validation set? What hyper-parameter settings were used?

L141: I don't understand how muscles were 'manually selected using thresholding'.

Section 2.5, fig 3.: Why were the initial U-Net predictions used to build the SSM? I believe that it would be better to use the original gold standard semi-automatic segmentations would lead to more accurate mean shapes. Now you are already limited by how good the initial U-Net predictions are. If this is not a mistake in writing, I believe this is quite crucial and requires more explanation.

L273: In other publications I have seen the RVE presented as an absolute value. Was it a conscious decision to not use the absolute volume difference?

Results:

L326: I fail to see how the Hausdorff distance becomes meaningless with larger objects. Yes the HD will likely increase compared to the analysis of the individual muscles, but it can still be used to compare the various post-processing methods, no?

Fig 7. I like this figure, but it's difficult to distinguish muscles where adjacent muscles are all colored dark blue.

Discussion:

Figure 10 and table 4 present results and thus fit better in the results section.

L460: 'General consistency between the cohorts' does not appear true to me. There may not be a statistical difference, but I can imagine that there is a difference in muscles between subjects with a BMI of 26.5 +- 3.4 and 22.0 +- 2.1. Did you check the difference in volume, shape, etc. between the muscles in PMW1 and PMW2?

It would be interesting to read your opinion on the generalizability of the MS post-processing method. It appears to me that you're confronted with quite a lot of smoothing when using a mean shape and this doesn't change when adding more subjects. Does this make this method only applicable for healthy, near-average subjects?

Reviewer #3: In this manuscript, authors present a new mean-shape based post-processing method in order to improve the automatic muscle segmentation accuracy through deep learning models, more specifically, UNet. The study is interesting and important because although there are many deep learning structures and algorithms having been proposed for the purpose of automatic image segmentation, these algorithms often lack sensitivity to details which hinders satisfactory semantic segmentation results. Post-processing is a good way to improve or enhance the segmentation accuracy.

1. The paper is very well organised and written. English may be improved.

2. You have compared your method with Mimics commercial semi-automatic muscle segmentation toolbox. What is the main purpose of this comparison? Does Mimic have post-processing?

3. There are lots of deep learning algorithms for image segmentation, why UNet?

4. In your comparison, Mimics has lower RVE, which is better than your method. Any consideration for future improvement of your algorithm?

5. It looks like mean shape is the mean value of this specific cohort. What about other cohort?

6. PLOS authors have the option to publish the peer review history of their article (what does this mean?). If published, this will include your full peer review and any attached files.

Reviewer #1: No

Reviewer #2: No

Reviewer #3: No

---

## [Author Response · Author response to Decision Letter 0]

15 Jul 2024

Reviewer 1 major comments.

1)

Q) The contribution of the study should be highlighted.

A) Thank you for your suggestions. The contribution of the study has been added in the end of the introduction section.

note) From Line 99 to 103, the contribution of this study has been highlighted: ‘The contributions of this study are: 1) A new mean shape (MS) based post-processing method is proposed, utilizing Statistical Shape Modelling (SSM) to fine-tune the segmentation output, taking into consideration the muscle anatomical shape; 2) A comparative study is conducted to investigate the effectiveness of post-processing approaches to the improvement of the DL automatic segmentation accuracy in terms of DSC, HD and ASSD.’

2)

Q) Obviously, the proposed post-processing approach can be applied in any segmented images. There are many different types of deep learning networks, can you explain why UNet was chosen as the deep learning model for this study?

A) The primary objective of this study is to investigate whether the proposed mean shape based post-processing method improves segmentation results. To achieve this, as a widely used segmentation model, UNet was chosen for training. Compared to other advanced models, UNet has fewer parameters, ensuring it does not excessively consume computational resources. Additionally, if the post-processing method proposed in this study proves to be effective, it can then be easily and directly applied to the output results of other studies that use UNet for training. This will facilitate direct comparisons between the original results and those obtained using our method.

3)

Q) In Participants, it looks like you used two different MRI scanners, were there any differences in imaging conditions between the two cohorts?

A) Although different scanners are used, the parameters used for imaging are the same. The imaging and post-processing operation details have been added in the revised draft.

note) From Line 112 to 123: ‘Although the two cohorts were imaged in separate studies, the scans were conducted at the same hospital using two different 1.5T MRI scanners, following the same scanning protocol. All complete lower-limb MRI scans were performed using the 1.5T Siemens Magnetom Avanto or the 1.5T Siemens Magnetom Aera (Siemens AG, Erlangen, Germany). Four sequences were acquired to image the hip, thigh, knee, and shank [21]. To reduce scanning time without compromising the detailed geometry of the joints, the joints were imaged at a higher resolution (pixel size of 1.05 mm × 1.05 mm and slice thickness of 3.00 mm) compared to the central portions of the long bones (pixel size of 1.15 mm × 1.15 mm and slice thickness of 5.00 mm). These sequences were then combined using MATLAB (R2006a) to create a continuous 3D image from the hip to the ankle. This was achieved by standardizing the resolution of each imaging sequence from the different sections through tri-linear interpolation (interp3, MATLAB R2006a) to 1.00 mm × 1.00 mm × 1.00 mm.’

4) 

Q) For the operators for manual segmentation, were they the same people or different people with similar technical level?

A) PMW-1 dataset was segmented by two PhD students and one postdoctoral researcher and PMW-2 dataset was segmented by one PhD student with similar technical level and trained on at least 15 datasets.

note) The sentence in section 2.1 from line 136 to 141 has been rephrased as follows: ‘The MR images of lower-limb muscles from PMW-1 were segmented by operators (two PhD students and one postdoctoral researcher) using the semi-automated muscle segmentation toolbox in Mimics 20.0 (Materialise, Leuven, Belgium), followed by some manual adjustments and post-processing [19]. The scans from PMW-2 were segmented by one operator (one PhD student) using the software Amira (version 2022.2; Thermofisher). All the operators have similar technical levels in terms of muscle segmentation and were trained on at least 15 datasets.’

5)

Q) For the training, one cohort of data (PMW-1) was used as training data while the second cohort was used for test. Is there any population bias in the cohort which may affect your result?

A) Both cohorts are composed of postmenopausal women, thus exhibiting similar physiological characteristics. As shown in Table 1, the average age difference is 2 years, and the average height is also similar. In comparative analysis, different methods are based on the results from the same samples (PMW-2), ensuring that the bias between PMW-1 and PMW-2 does not significantly impact this study's outcomes. Section 3.1 and the appendix of this paper provide a detailed analysis of the PMW-2 individuals (S2 Fig). 

6) 

Q) In the proposed method, mean shape of the cohort (postmenopausal women) was used. What is your view for the method to apply to different population, say young people?

A) Compared to postmenopausal women, MRI muscle slices of younger individuals exhibit different textural characteristics. For example, due to ageing, muscle atrophy in postmenopausal women leads to less compact muscle boundaries (as shown in Fig. R1), facilitating better segmentation by DL models. In contrast, younger individuals have stronger muscles with potentially less distinguishable boundaries between adjacent muscles. Due to muscle loss with age, some soft tissues (such as fibers and ligaments) are relatively obvious, showing low signals on the T1 sequence and showing black textures in the images.Therefore, the output results of deep learning models are expected to vary across different cohorts.

Fig. R1 MRI slice comparison between different cohort (seen in file, 'response to reviewers')

Regarding the proposed post-processing methods, this study focuses more on geometric morphological changes. If the pre-generated mean shape of a particular muscle closely resembles the shape and volume of the muscle in a new sample, the post-processing method would be effective. However, if there are significant differences in muscle volume, length, and morphology between the different cohorts, further research will consider generating different mean shape dataset for different cohorts to appropriately classify and process the incoming samples.

Reviewer 2 major comments.

1)

Q) When data is pooled, is this done after the predictions? Or is a separate U-Net trained for the pooled data. If only 1 label is available for training the U-Net I imagine that you could achieve much better results compared to only pooling in post-processing.

A) Yes, the pooling operation is done after the prediction. If a model is used to learn and predict a specific muscle (binary segmentation task), it is likely to yield more accurate results. However, this approach would require significantly more training time and computing consumption. In this study, we employ pooling to observe the differences in the recognition abilities of various methods on muscle regions at a macro level.

2)

Q) Just reading the introduction, the goals of the study slightly confuse me. In L91 you mention that a new SSM-based post-processing method is proposed in this study. However, in the subsequent paragraph, you say that the goal of the study is to assess the effect of existing post-processing methods, with no mention of the proposed novel method. From the rest of the paper, it is clear that this SSM-based method is an essential part of the goal of this study.

A) The sentence has been rephrased and added to introduction from line 95 to 103 as follows:

‘The goal of this study was to propose a new mean shape based post-processing method and to assess the effect of it with existing methods (CRF, and 2D post-processing method) on the accuracy of a previously developed CNN approach to segment individual muscles of the lower limb from MRI scans [4]. The secondary goal of the study was to compare the accuracy of different segmentation tools, including that of a commercial semi-automatic muscle segmentation toolbox (Mimics 26.0, Materialise). The contributions of this study are: 1) A new mean shape (MS) based post-processing method is proposed, utilizing Statistical Shape Modelling (SSM) to fine-tune the segmentation output, taking into consideration the muscle anatomical shape; 2) A comparative study is conducted to investigate the effectiveness of post-processing approaches to the improvement of the DL automatic segmentation accuracy in terms of DSC, HD and ASSD.’

3)

Q) More information regarding the MR images should be presented (pixel size, slice thickness, slice increment, FOV, any filtering applied)

A) The detailed information of the MR images has been added to the section 2.1 from line 112 to 123 as follows.

note) From Line 112 to 123: ‘Although the two cohorts were imaged in separate studies, the scans were conducted at the same hospital using two different 1.5T MRI scanners, following the same scanning protocol. All complete lower-limb MRI scans were performed using the 1.5T Siemens Magnetom Avanto or the 1.5T Siemens Magnetom Aera (Siemens AG, Erlangen, Germany). Four sequences were acquired to image the hip, thigh, knee, and shank [21]. To reduce scanning time without compromising the detailed geometry of the joints, the joints were imaged at a higher resolution (pixel size of 1.05 mm × 1.05 mm and slice thickness of 3.00 mm) compared to the central portions of the long bones (pixel size of 1.15 mm × 1.15 mm and slice thickness of 5.00 mm). These sequences were then combined using MATLAB (R2006a) to create a continuous 3D image from the hip to the ankle. This was achieved by standardizing the resolution of each imaging sequence from the different sections through tri-linear interpolation (interp3, MATLAB R2006a) to 1.00 mm × 1.00 mm × 1.00 mm.’

4)

Q) L112-L124: it is unclear to me if the segmentations of PMW-1 were performed as part of this study, or in another study (reference 19). If I understand correctly, the scans for PMW-2 were segmented for this specific study. Were exactly the same muscles extracted? How was the segmentation performed? Was this also performed semi-automatically and performed by multiple operators and then compared? I believe that these segmentations are considered the golden standard mentioned in the results. I suggest mentioning this explicitly for clarity.

A) Yes, PMW-1 was manually segmented in another study [ref 19], and PMW-2 was manually segmented in this study. The same muscles were extracted. 

note) The sentence in section 2.1 has been rephrased from line 136 to 141 as follows:

‘The MR images of lower-limb muscles from PMW-1 were segmented by operators (two PhD students and one postdoctoral researcher) using the semi-automated muscle segmentation toolbox in Mimics 20.0 (Materialise, Leuven, Belgium), followed by some manual adjustments [19]. The scans from PMW-2 were segmented by one operator (one PhD student) using the software Amira (version 2022.2; Thermofisher). In the software, the operator manually selects the target region on each MR slice to perform purely manual segmentation. All the operators have similar technical levels in terms of muscle segmentation and were trained on at least 15 datasets.’

5)

Q) I suggest trying to put table 2 into a flowchart figure as is often done with U-Net networks.

A) Following the practice outlined in reference 6, the layers parameters of U-Net are described in more detail in table to facilitate comparison and can provide a reference for readers. Following the suggestion, a flowchart figure of the U-Net structure is added in supporting files (S1 Fig).

6)

Q) L131: I'm a bit confused regarding how the U-Net was trained. Why was the U-Net trained and tested 3 times? Was it retrained for each method of post-processing? If so, wouldn't it make more sense to perform the post-processing on one trained model? Was there no validation set? What hyper-parameter settings were used?

A) The purpose of the repeated training and testing is to provide a means for statistical analysis, which will be able to evaluate the effectiveness of the methods. In this study, we chose to repeat 3 times. Each time was followed by applying various post-processing methods to the results. The purpose of this study is not to select the best model architecture; therefore, no special validation set was reserved. During the training period, the dataset was randomly split into training and validation sets with a 9:1 ratio. The initial learning rate for the model was set to 0.001, and it decayed by 0.9 every 5 epochs, for a total of 100 epochs. The model was trained with the multi-class cross entropy loss function. Other hyper-parameters were tuned empirically for the optimal network performance, whilst ensuring the GPU memory and capacity were not exceeded. After the loss converged on the training set, the best-performing set of weights from each epoch on the validation set was retained.

note) The training detail information has been added to section 2.2 from line 147 to 152 as follows: ‘During the training period, the dataset was randomly split into training and validation sets with a 9:1 ratio. The initial learning rate for the model was set to 0.001, and it decayed by 0.9 every 5 epochs, for a total of 100 epochs. The model was trained with the multi-class cross entropy loss function. Other hyper-parameters were tuned empirically for the optimal network performance, whilst ensuring the GPU memory and capacity were not exceeded. After the loss converged on the training set, the best-performing set of weights from each epoch on the validation set was retained.’

7) 

Q) L141: I don't understand how muscles were 'manually selected using thresholding'.

A) In the process of using Mimics, the software requires users to select the approximate region of the muscle in the MR slices as a mask without needing high precision. The thresholding method is one option provided by the software. Given the significant contrast between the muscle region and the surrounding tissues (e.g. fat tissue) in MRI slices, the thresholding method can conveniently select the approximate muscle area. During usage, the threshold is manually adjusted by observing the coverage of the muscle mask. The manual adjustment continues until the mask includes the approximate muscle region without covering excessive noise.

8)

Q) Section 2.5, fig 3.: Why were the initial U-Net predictions used to build the SSM? I believe that it would be better to use the original gold standard semi-automatic segmentations would lead to more accurate mean shapes. Now you are already limited by how good the initial U-Net predictions are. If this is not a mistake in writing, I believe this is quite crucial and requires more explanation.

A) The mean shape is derived from the precise manual segmentation results (gold standard) of PMW-1. This clarification has been further elaborated in the revised manuscript.

note) The sentence has been revised and added to section 2.3 from line 207 to 208 as below: ‘Using the ShapeWorks software, the mean shape for each muscle was generated (a-2) using gold standard manual segmentation from PMW-1.’

9)

Q) L273: In other publications I have seen the RVE presented as an absolute value. Was it a conscious decision to not use the absolute volume difference?

A) The reason for not using absolute values is to understand the specific differences in muscle volume prediction among different methods, whether they tend to overestimate or underestimate. If biases exhibit systematic consistency, adjustments can be made manually during practical application to align the results more closely with actual values.

10)

Q) L326: I fail to see how the Hausdorff distance becomes meaningless with larger objects. Yes, the HD will likely increase compared to the analysis of the individual muscles, but it can still be used to compare the various post-processing methods, no?

A)

Original content: ‘ When the labels of the 16 target muscles were pooled into one label, DSC, RVE, and ASSD were used to evaluate the recognition performance of each method on the overall target muscle region (Fig 5). Because the whole muscle cohort is larger and longer than the individual muscle, the comparison of HD is meaningless and therefore removed.’

For medical applications, measuring muscle volume is a focal point of researchers' att

---

## [Decision Letter · Decision Letter 1]

29 Jul 2024

A novel mean shape based post-processing method for enhancing deep learning lower-limb muscle segmentation accuracy

PONE-D-24-19418R1

Dear Prof. Guo,

We’re pleased to inform you that your manuscript has been judged scientifically suitable for publication and will be formally accepted for publication once it meets all outstanding technical requirements.

Kind regards,

Fei Yan

Academic Editor

PLOS ONE

Additional Editor Comments (optional):

Reviewers' comments:

Reviewer's Responses to Questions

**Comments to the Author**

1. If the authors have adequately addressed your comments raised in a previous round of review and you feel that this manuscript is now acceptable for publication, you may indicate that here to bypass the “Comments to the Author” section, enter your conflict of interest statement in the “Confidential to Editor” section, and submit your "Accept" recommendation.

Reviewer #1: All comments have been addressed

Reviewer #2: All comments have been addressed

Reviewer #3: All comments have been addressed

2. Is the manuscript technically sound, and do the data support the conclusions?

Reviewer #1: (No Response)

Reviewer #2: Yes

Reviewer #3: Yes

3. Has the statistical analysis been performed appropriately and rigorously? 

Reviewer #1: (No Response)

Reviewer #2: Yes

Reviewer #3: Yes

4. Have the authors made all data underlying the findings in their manuscript fully available?

Reviewer #1: (No Response)

Reviewer #2: No

Reviewer #3: Yes

5. Is the manuscript presented in an intelligible fashion and written in standard English?

Reviewer #1: (No Response)

Reviewer #2: Yes

Reviewer #3: Yes

6. Review Comments to the Author

Reviewer #1: (No Response)

Reviewer #2: (No Response)

Reviewer #3: (No Response)

7. PLOS authors have the option to publish the peer review history of their article (what does this mean?). If published, this will include your full peer review and any attached files.

Reviewer #1: **Yes: **Quanmin Zhu

Reviewer #2: No

Reviewer #3: No

---

## [Editor Report · Acceptance letter]

25 Sep 2024

PONE-D-24-19418R1 

PLOS ONE

Dear Dr. Guo, 

I'm pleased to inform you that your manuscript has been deemed suitable for publication in PLOS ONE. Congratulations! Your manuscript is now being handed over to our production team.

Kind regards, 

on behalf of

Dr. Fei Yan 

Academic Editor

PLOS ONE